# Algal Oxylipins Mediate the Resistance of Diatoms against Algicidal Bacteria

**DOI:** 10.3390/md16120486

**Published:** 2018-12-04

**Authors:** Nils Meyer, Johanna Rettner, Markus Werner, Oliver Werz, Georg Pohnert

**Affiliations:** 1Institute for Inorganic and Analytical Chemistry, Bioorganic Analytics, Friedrich Schiller University Jena, Lessingstr. 8, D-07743 Jena, Germany; Nils.meyer@uni-jena.de (N.M.); johanna.rettner@uni-jena.de (J.R.); 2Department for Pharmaceutical/Medicinal Chemistry, Institute of Pharmacy, Friedrich Schiller University Jena, Philosophenweg 14, D-07743 Jena, Germany; werner.markus@uni-jena.de (M.W.); Oliver.werz@uni-jena.de (O.W.)

**Keywords:** HEPE hydroxylated eicosapentaenoic acid, HETE hydroxylated eicosatetraenoic acid, oxylipins, diatoms, plankton, algicidal bacteria, induced chemical defense, resolvins

## Abstract

Algicidal bacteria can lyse microalgal blooms and trigger shifts within plankton communities. Resistant algal species can escape lysis, and have the opportunity to dominate the phytoplankton after a bacterial infection. Despite their important function in ecosystem regulation, little is known about mechanisms of resistance. Here, we show that the diatom *Chaetoceros didymus* releases eicosanoid oxylipins into the medium, and that the lytic algicidal bacterium, *Kordia algicida*, induces the production of several wound-activated oxylipins in this resistant diatom. Neither releases nor an induction occurs in the susceptible diatom *Skeletonema costatum* that is lysed by the bacterium within a few days. Among the upregulated oxylipins, hydroxylated eicosapentaenoic acids (HEPEs) dominate. However, also, resolvins, known lipid mediators in mammals, increase upon exposure of the algae to the algicidal bacteria. The prevailing hydroxylated fatty acid, 15-HEPE, significantly inhibits growth of *K. algicida* at a concentration of approximately 1 µM. The oxylipin production may represent an independent line of defense of the resistant alga, acting in addition to the previously reported upregulation of proteases.

## 1. Introduction

Multiple processes govern the complex species succession in plankton. It has been recognized, early on, that nutrient availability and abiotic conditions favor certain species that can reach dominating abundance within the community [1]. Later, also chemical mediators that can inhibit or promote cell proliferation and regulate predator-prey or host-pathogen interactions were recognized as additional factors that govern the complexity of plankton [2,3]. Precise regulation is a hallmark of such chemically mediated interactions. Production of defensive chemicals can be induced upon recognition of signals from the attacker, or activated by wounding that occurs in the interaction situation [4,5]. Diatoms that frequently dominate phytoplankton communities have evolved mechanisms to counter predation by the wound-activated formation of polyunsaturated aldehydes that inhibit the proliferation of herbivorous copepods [6,7]. These responses are triggered by cell disruption, that leads to rapid activation of lipases and lipoxygenases, building up massive local amounts of defensive aldehydes [8,9]. Further higher molecular weight oxylipins have been reported from diatoms, and are potentially involved in wound-activated defense, however, their ecological role is not yet fully understood [10,11]. These higher oxylipins include hydroxylated and further oxygenated metabolites, mainly derived from eicosapentaenoic acid and docosahexaenoic acid [12]. In contrast to wound-activated processes, induced defenses do not require cell disruption but, rather, the perception of signals from an attacker or pathogen, that then triggers an often slower de novo biosynthesis of defensive metabolites. Such elaborate mechanisms that require signal recognition and subsequent regulation of metabolic pathways are, however, only very rarely reported from microalgae [13].

Besides herbivory, infection by pathogens is a second major cause of death for unicellular algae in water. Viral infections and algicidal bacteria play major roles in the control of populations, or even the termination of entire blooms [14,15]. Algicidal bacteria can substantially reduce native diatom populations, and are also utilized in biotechnology for cell lysis. Their mode of action is as diverse as is their host specificity [16]. Both generalists, as well as specialists, have been identified. The here-investigated *K. algicida*, that was isolated from a *S. costatum* bloom [17], is rather universal at lysing diatoms, with few identified exceptions. An example of a resistant alga is the ubiquitous diatom *Chaetoceros didymus*, that survives even incubations with high bacterial densities [18]. One response of *C. didymus* to the lytic bacteria is the upregulation of proteases that have the potential to counteract the lytic enzymes from bacteria [18]. Here, we hypothesize that comparison of the response of resistant and non-resistant diatoms allows deducing of information on the mechanisms of infection and resistance. We introduce a targeted lipidomics study surveying the endo- and exometabolome of the resistant *C. didymus*, and the susceptible diatom *Skeletonema costatum*, for metabolites that are involved in resistance mechanisms. We identify an induced upregulation of a family of oxylipins in the wound-activated response of the resistant alga, as well as a release of these metabolites, that may contribute to the chemical defense against bacteria.

## 2. Results and Discussion

Exponentially growing algal cultures were incubated with a suspension of *K. algicida* at a final OD_550_ of 0.01. These conditions were sufficient to lyse *S. costatum* within three days, while *C. didymus* was resistant and grew with similar kinetics as the control (Figure 1). This pattern is consistent with previous investigations addressing the infection of diatoms with *K. algicida* [19]. By incubating with a bacterial density of OD 0.01, we achieved a sufficiently slow lysis of *S. costatum*, allowing for examination of potential induced responses. 

To investigate the chemistry that is potentially mediating bacterial/algal interactions within the water surrounding the producing cells, we initially focused on metabolites released into the medium. Therefore, we adapted a lipidomics approach, initially developed to survey C18 to C22 fatty acid-derived oxylipins from diatoms [10]. After centrifugation of the cultures (1700× *g*, 15 min, 4 °C), a careful filtration (pore size 1.2 µm) of the supernatant was conducted on glass fiber filters to avoid cell disruption. Extraction of the resulting filtrate with reversed phase C18 solid phase extraction cartridges gave samples for liquid chromatography/mass spectrometry (LC/MS). Quantifiable amounts of oxylipins in the pg mL^−1^ to ng mL^−1^ range were detected. Given the fact that algae build up elevated concentrations of primary and secondary metabolites in the phycosphere (a diffusion-limited zone around the cells), the local concentrations surrounding the cells might be substantially higher [20]. In the medium of the resistant *C. didymus*, a clear accumulation of extracellular oxylipins was observed, with a concomitant increase of free EPA (Figure 2). This increase occurred in infected, as well as non-infected, *C. didymus*. The majority of the detected oxylipins belong to the family of hydroxylated eicosapentaenoic acids (HEPEs). The most abundant member of this family was 15-HEPE, followed by 12-, 18-, 11-, and 5-HEPE. Concentrations of 15-HEPE in the medium reached ca. 28 ng mL^−1^. Even if this value is below the activity threshold for the inhibition of *K. algicida* growth (see below) it might very well contribute to the chemical defense of the alga. If we consider that 28 ng mL^−1^ represents an average concentration in the medium, the local concentrations in the phycosphere of a producing cell will exceed this value substantially [20]. Active concentrations could be reached in the distance of few micrometers around the producing alga and, thus, within the scale of the bacterial cell. Only minor differences in the level of released oxylipins were observed between uninfected controls and *K. algicida*-infected treatments (Figure 2). These differences were significant only for 5-HEPE on day 4 (*p* = 0.027). *C. didymus* is, thus, constitutively building up elevated amounts of the oxylipins in its phycosphere. To our knowledge, this is the first time that release of higher molecular weight oxylipins from non-disrupted diatoms is reported. The kinetics of oxylipin accumulation exceeds that of the growth, except for RvE_1_ and, as a result, oxylipin release rates normalized to chlorophyll *a* fluorescence increase over time (Figure 1 and Figure 2).

Enzymatic formation of the observed hydroxylated fatty acids requires the action of lipoxygenases that insert oxygen in a position-specific manner into the substrate EPA [21]. The observed positional specificity (Figure 2) confirms that product formation in *C. didymus* is occurring enzymatically, and not by abiotic oxidation of the polyunsaturated fatty acids. The initial enzymatic oxidation is followed by further reduction of the intermittently formed hydroperoxides to the corresponding alcohols [22]. At this stage, it cannot be concluded whether the oxylipins are formed intracellularly, or in the phycosphere, and how the metabolites or their precursors are released out of the cells. *Skeletonema marinoi* releases short chain oxylipins (polyunsaturated C6 and C8 aldehydes) into the seawater just before the beginning of the declining growth phase, but no release of these and higher oxylipins was observed during the early phases of growth as it is considered here [23]. In the spent medium of *S. costatum* none of the higher molecular weight oxylipins, investigated in this study exceeded concentrations of 50 pg mL^−1^, and no trend throughout the growth, in the presence or absence of bacteria, was observed (data not shown). *S. costatum* releases volatile oxylipins, as well as HEPEs, upon cell disruption using lipoxygenase-mediated pathways [10,24,25]. However, cell disruption, as it occurs during *K. algicida* infection, does not lead to an increased formation of HEPEs (data not shown). This might be due to a suppression of the lipase/lyase pathways through the bacteria, or due to the metabolization of the algal oxylipins. The very low concentrations of oxylipins found in *S. costatum* extracts might not necessarily stem from enzymatic transformations, but can arise, alternatively, from abiotic oxidation of free fatty acids.

The underlying principles of oxylipin formation in *C. didymus* were further investigated by monitoring the capacity of the cells to produce these metabolites after wound activation. In agreement with a previous study, hydroxylated fatty acids were detected in amounts of up to 100 fg cell^−1^ in the resistant alga after lysis of the cells (Figure 3) [10]. Detected oxylipins included HEPEs, resolvins (Rvs), hydroxylated docosahexaenoic acids (HDHAs), hydroxylated octadecadienoic acids, and hydroxylated eicosatetraenoic acids (HETEs). The level of HEPEs were higher in those algae that were previously challenged with *K. algicida* (Figure 3A). RvE_1_ (a tri-hydroxylated EPA-derived oxylipin) was not detected after wound activation, but the di-hydroxylated RvE_3_ was significantly increased in bacterially infected cells (Figure 3B).

To verify if the release and the wound-activated formation of oxylipins constitutes a defense mechanism of the alga against the bacterial pathogen, we evaluated the effect of the most abundant oxylipin, 15-HEPE, on the bacteria. This oxylipin inhibits the growth of the bacteria for 6 h in a concentration-dependent manner, with an effective concentration of 1 µg mL^−1^ (Figure 4). The lysis event of a single diatom cell results in the release of 15-HEPE into a small volume surrounding the cell, barely exceeding the cellular dimensions. If cellular production is normalized to the reported cell volume of 2200–7600 µm^3^, the concentrations after lysis would exceed the effective concentration more than 10-fold [26,27]. Accordingly, oxylipins can contribute to a chemical defense of the alga. As discussed previously, for phytoplankton, in general, such wound-activated defense of an individual cell would benefit the population of closely related conspecifics [2,28]. We propose the C20 oxylipins could represent a line of defense against *K. algicida* that acts in addition to the previously reported defensive proteases [18].

We, here, extend the concept of induced oxylipin production to the field of C20 oxylipins. We suggest that these metabolites might be involved in the chemical defense of diatoms. While this orchestration of signaling is new to unicellular algae, comparable processes of differential response to wounding and induction have been reported frequently from higher plants [29]. There, the lipoxygenase metabolite jasmonic acid is a key regulator for most diverse induced defensive responses. Such a “master switch” is not known for microalgae, and it has to be explored if these unicellular algae require such elaborate hormones.

## 3. Materials and Methods

### 3.1. Algal Cultivation and Quantification

*C. didymus* was isolated by W. Kooistra from the Mediterranean Sea (Stazione Zoologica Anton Dohrn, Naples, Italy), and is maintained in our culture collection; *S. costatum* (RCC75) was obtained from the Roscoff Culture Collection (Roscoff, France). Algae were grown in batch cultures according to published procedures [18,30] under a 14/10 h light/dark regime with an illumination of 15–22 µmol photons m^−2^ s^−1^.

For determination of growth, in vivo chlorophyll *a* fluorescence was measured in technical triplicates on a Mithras LB 940 plate reader (Berthold Technologies, Bad Wildbad, Germany).

For microalgal quantification, Lugol-fixed cells were counted under a LEICA DM 2000 light microscope in a Nannoplankton Chamber (PhycoTech Inc., St. Joseph, MI, USA; for *C. didymus*) or a Fuchs-Rosenthal counting chamber (for *S. costatum*). The cell counts of the cultures were corrected using the counts of non-precipitated cells in the supernatant after centrifugation.

### 3.2. Bacterial Cultivation and Quantification

For co-cultivation experiments, *K. algicida* OT-1 [17] was spread on marine agar and cultivated at 23 °C for 2–3 days. The bacterial lawn was removed with a sterile cotton swab, and resuspended in algal culturing medium to an OD_550_ of 0.5 determined on a Genesys 10S UV–Vis spectrophotometer (Thermo Fisher Scientific, Waltham, MA, USA).

For recording the dose–response curve, *K. algicida* was cultivated in marine broth medium at 23 °C and with constant agitation on a shaker at 78 rpm. An overnight culture was diluted with marine broth to an OD_550_ of 0.5.

### 3.3. Co-Cultivation Experiment

For extracellular oxylipin profiling, exponentially growing algal cultures (245 mL each) were inoculated with *K. algicida* suspension (5 mL), in triplicates, to a final OD_550_ of 0.01 (treatment), or with the same amount of algal culturing medium (control). Samples for cell counts and oxylipin profiling were taken daily for 4 days.

For oxylipin profiling of wound-activated cells, cultures of *C. didymus* (54 mL each) were inoculated with *K. algicida* suspensions (1.1 mL), in 7 replicates, to a final OD_550_ of 0.01 (treatment), or with the same amount of algal culturing medium (control). After 4 days, samples for cell counts and oxylipin profiling were taken.

### 3.4. Wound Activation and Oxylipin Profiling

The procedure is described in detail elsewhere [10]. Briefly, algal cells are lysed using an ultrasonic probe, incubated at room temperature for 10 min, and quenched by addition of methanol and internal standard. Oxylipins are extracted using C_18_ SPE columns, and analyzed on an Acquity UPLC (Waters, Milford, MA, USA) equipped with a BEH C_18_ column (Waters) coupled to an AB Sciex 5500 QTrap (Sciex, Toronto, ON, Canada) mass spectrometer with ESI source in negative mode.

### 3.5. Extracellular Oxylipin Profiling

After centrifugation (1700× *g*, 15 min, 4 °C) of algal cultures, the supernatant was 1.2 µm-filtered on a glass fiber filter (Whatman^TM^ GF/C, GE Healthcare, Little Chalfont, UK). Flow-through (14 mL) was added to 28 mL cooled methanol and internal standard. The samples were processed, as described, for the oxylipin profiling after wound activation (see above), with volumes adjusted.

### 3.6. Determination of Concentration-Dependent 15-HEPE Activity Against K. algicida

In 96-well plates, 5 µL aliquots of *K. algicida* culture were added to 15-HEPE (Cayman Chemical, Ann Arbor, MI, USA) in 195 µL marine broth medium at concentrations of 0, 0.03, 0.1, 0.3, 1, 3, 10, and 30 µg/mL in 5 replicates. Plates are sealed with Parafilm^®^ (Pechiney Plastic Packaging, Chicago, Il, USA), and covered with aluminum foil. At these conditions, *K. algicida* was cultured for 32 h, and in intervals of 2 h, the plate was shaken for 10 s (Ø 2 mm, double orbit) and the OD_550_ was measured (0.1 s, lamp energy 10,000) using a Mithras LB 940 plate reader (Berthold Technologies, Bad Wildbad, Germany). The slope of *K. algicida* growth in the treatments is assessed relative to the slope in the negative control.

## Figures and Tables

**Figure 1 marinedrugs-16-00486-f001:**
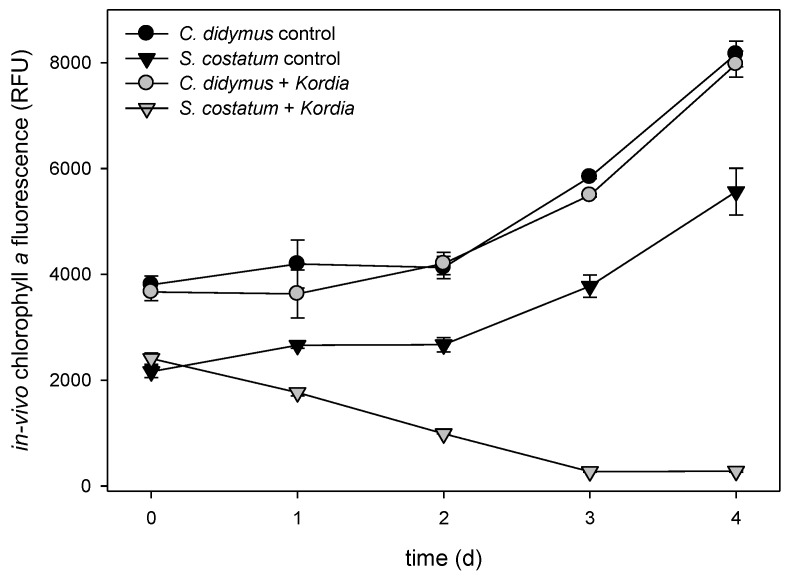
Growth of *Chaetoceros didymus* and *Skeletonema costatum* (mean ± SD, *n* = 3) monitored as in vivo chlorophyll *a* fluorescence. Values are given in relative fluorescence units (RFU), over time, for control cultures and cultures treated with *Kordia algicida*.

**Figure 2 marinedrugs-16-00486-f002:**
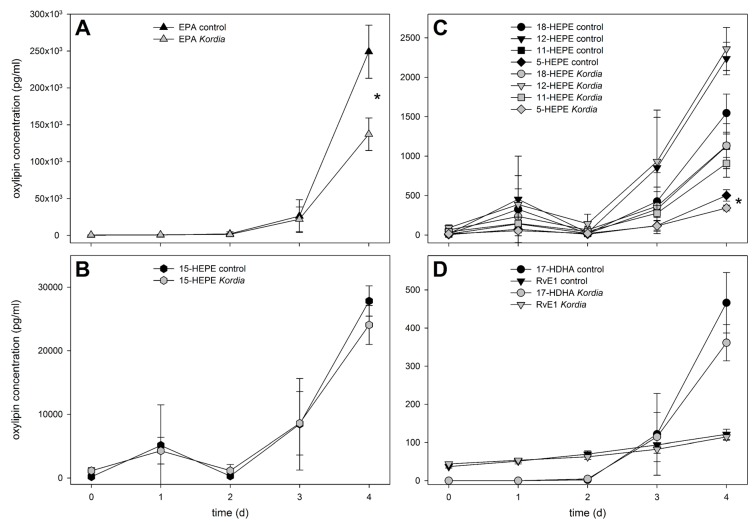
Extracellular concentrations of oxylipins and free eicosapentaenoic acid (EPA) in *C. didymus* culture medium (mean ± SD, *n* = 3). Panels depict EPA (**A**), hydroxylated eicosapentaenoic acids (HEPEs), namely 15-HEPE (**B**) and 18-, 12-, 11-, and 5-HEPE (**C**), as well as 17-hydroxydocosahexanoic acid (17-HDHA) and resolvin (Rv)E_1_ (**D**). Statistically significant differences (unpaired *t*-test) at day 4 between control cultures and cultures treated with *Kordia algicida* are marked with * (*p* < 0.05).

**Figure 3 marinedrugs-16-00486-f003:**
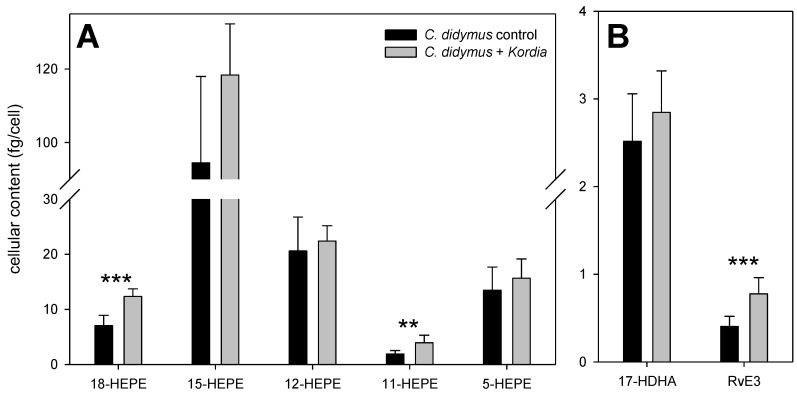
Oxylipins detected 10 min after cell disruption of *C. didymus*. Depicted is the cellular production (mean ± SD, *n* = 7) of 5 different hydroxylated eicosapentaenoic acids (HEPEs) (**A**), as well as 17-hydroxydocosahexaenoic acid (HDHA) and resolvin (Rv)E_3_ (**B**) after wounding of *C. didymus* at day 4. Statistically significant differences (unpaired *t*-test) between control cultures and cultures treated with *K. algicida* are marked with ** (*p* < 0.01) and *** (*p* < 0.001).

**Figure 4 marinedrugs-16-00486-f004:**
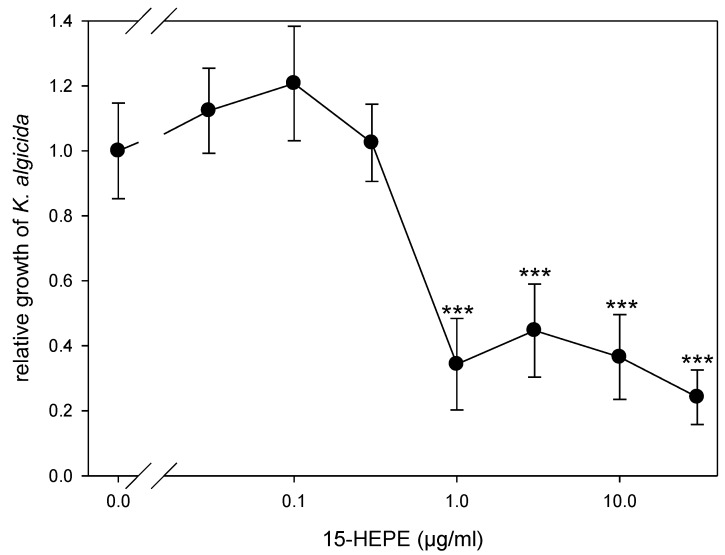
Concentration–response curve for 15-HEPE inhibition of *Kordia algicida* growth (mean ± SD, *n* = 5). Growth was monitored after 6 h in the presence of different concentrations of 15-hydroxylated eicosapentaenoic acid (HEPE) and normalized to the control at 0 µg/mL (no added HEPE). Statistically significant differences (unpaired *t*-test) between control cultures and cultures treated with *K. algicida* are marked with *** (*p* < 0.001).

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
