# Peer review of "Algal Oxylipins Mediate the Resistance of Diatoms against Algicidal Bacteria"

_marinedrugs, 2018, doi:10.3390/md16120486_

Reviewer 1 Report

This manuscript "Induced algal oxylipins mediate the resistance of diatoms against algicidal bacteria" is submitted as an article in Marine Drugs.

 The authors claimed to have identified oxylipins responsible of the resistance of C. didymus after bacterial infection with K. algicida. The authors described the production of oxylipins as induced by the bacterial infection.

The overall data are not so obvious, in Fig 2 only 5-HEPE seems to be induced but it is not the case in the Fig 3 after cell disruption. In this later figure, only the HEPE derivatives (18-HEPE and 11-HEPE but also RvE3) in very small amount appear to be modulated by the presence of bacteria. There is no evidence that these compounds were particularly active ones, the activity being tested on the major compound 15-HEPE.

 The evidence of 15-HEPE activity is also questionable. The estimate of the intracellular concentration is OK but is not related with the expected extracellular activity. Furthermore, a bacterial resistance is proposed to be acquired in 6 hours and is not compatible with a protection of C. didymus over a 4 days period. A proposed explanation was the protease release from C. didymus Ref 26. In this reference, these proteases induced after 2 days might have an (additional) adverse effect on S. costatum.

The use of short term medium of C. didymus induced or not with K. algicida (before induction of proteases) should help the growing of S. costatum in presence of bacteria. This could constitute a good probe of the activity of the metabolites release by C. didymus. But it may be difficult to manage such experiment.

Minor corrections:

L 104 The data concerning S. costatum are announced to be presented in the Fig 2 but absent!

The amount of 17-HDHA seems different in Fig 3 when compared to Table 1 in Ref 10 (# 2,5 versus 1.86)

Author Response

Reviwer 1 (responses in bold)

The manuscript by Meyer et al. reports on studies of algal/diatom chemical ecology mediated by EPA-derived fatty acids. The work draws its conclusion from unsupported assumptions and weak correlations and needs significant re-focusing or additional supporting data.

A major concern is the leap from correlations of concentration variability to statements of causation. So three of seven (less than half) oxylipins (Fig 3) show statistically significant increase in bacterial-challenged diatoms relative to control diatoms. For those three, there is clearly a correlative effect, but no experimental evidence is presented as to what that effect is – defensive, offensive, or perhaps an unrelated downstream response. Instead, the authors take a metabolite that does not respond to the bacterial challenge and show that at concentrations (1 uM) well above what is found in the media (28 ng/mL = 88 nM)), the bacterial growth is slowed for six hours (but not longer). There is some discussion (lines 127-130) about phycosphere concentrations of the lipids and how they would exceed the effective concentration, but that's entirely conjecture and completely unknown. Even the authors state (line 126,127) '...it might very well contribute to the chemical defense...'. However, they move on into the discussion assuming it does, concluding 'The oxylipin therefore constitutes a first line of defense against...' the bacterium. That conclusion is not warranted based on the data provided.

 We see the argument of the reviewer and describe our results now in a more reflected manner. As consequence we also adjusted the title and removed “Induced”.  From the title onwards the manuscript was adjusted so that no wrong suggestions are made. We agree that he phycosphere discussion was misplaced and moved it to the discussion of the released metabolites (now line111).

Minor concerns:

line 20: 15-HEPE does not quantitatively inhibit K. algicida. It significantly inhibits it, but the relative growth (Fig 4) never falls below 0.2.

changed to “significantly”

line 21-22: conclusion overstated

now toned down “The oxylipin release may represent an…”

line 39: the term 'connected' has little scientific value. Do oxylipins cause wound-activated defenses or do they merely vary in response – those are both 'connections' but represent significantly different ecological implications

re-phrased “discussed as potential defense”

line 55-56: A discussion of how K. algicida, C. didymus and S. costatum interact in nature is important. Are they sympatric? Naive? And whatever they are, how does that matter for this study?

We now mention “K. algicida that was isolated from a S. costatum bloom [17]“, add ref. 17 and mention that C. didymus is ubiquitous. We also explain “ Comparison of the response of resistant and non-resistant diatoms allows deducing information on the mechanisms of infection and resistance.”

line 58: conclusion overstated

now “may contribute”

line 61: 'quantitatively' too strong a word

deleted

line 62: 'entirely' too strong

deleted

line 90: 'exceeds that of the growth' (drop 'curve')

Adjusted to “growth”

Line 91: there is no growth curve in Fig 1 and 2

Now not referring to growth curve but to “normalized chlorophyll a fluorescence”

line 104: Fig 2 has no S. costatum data

Corrected  “data not shown”

line 107: 'but can arise alternatively by the abiotic oxidation of free fatty acids'. If this is true for S. constatum, why is this not true of C. didymus?

Despite the fact that the precursor content (polyunsaturated fatty acids) is similar in both algae the levels of oxylipins in C. didymus is clearly higher. Since in addition culture conditions were identical, this abiotic pathway would only contribute to a very minor extend. Further, the positional specificity (predominant formation of 15-HEPE) clearly contradicts abiotic formation that would not be as specific. This is now discussed in line 132ff.

line 120: 'Consistently' is a weak scientific word. The level of HEPEs and 17-HDHA is not significant, so there is no correlation. No significance, no 'consistently'.

corrected

line 122: 12-HEPE did not increase significantly – the statement is wrong

We mention now that the increase is not significant.

line 124: RvE3 increased 'significantly' at the attamolar level – that needs to be put in context. Figure 3B should be part of Fig 3A with everything on the same scale. In the absence of bioassay data, an attamolar level increase in concentration is meaningless

There might be a misunderstanding: given a cell volume of ca. 5000 µm3 fg/cell amounts would correspond to micromolar concentrations. Especially since these compounds are highly active with clear specificity (in mammals several orders of magnitude lie between the activity of resolvins and HEPEs) we want to leave the graph as it is to document the regulation without pre-judgement of a potential activity.

line 132: broken graph unjustifiable. Put everything on scale.

If put to scale18-, 11-, and even 5-HEPE signals would not be resolved enough to recognize the error bar – we prefer to leave the break.

line 139: that term 'connected' again. Not scientifically meaningful (see line 39 comment)

Rephrased: “To verify if  the release of oxylipins constitutes a defense mechanism of the alga against the bacterial pathogen,…”

line 140: 15-HEPE was not significantly modulated by treatment with bacterial pathogen

We refer to the “most abundant oxylipin” – this is true even if the observed up-regulation is not significant.

line 143-144: intracellular concentrations are irrelevant because the bacterial pathogen never experiences intracellular conditions

We did not want to suggest that bacteria enter the cells. We thus re-phrased: “In light of an activated defensive mechanism also the lysis event of a single diatom cell has to be taken into account, which would release the intracellular content of 15-HEPE into a small volume not much exceeding the cellular dimensions. If natural amounts from the cells are normalized to the reported cell volume of 2200 µm3 – 7600 µm3, the concentrations after lysis would exceed the effective concentration always more than 10-fold [25]. In a monoclonal population this lysis of an individual would benefit the population [26].”

line 145: 'connected' again

Rephrased: “According to the results presented here, oxylipins can contribute to a chemical defense in the alga”

line 147: authors state 'This might be seen as an adaptation mechanism...' then conclude (line 149-150) that it is, without any support. Jumping from '...might be seen...' to '...therefore constitutes a first line of defense...' is unjustifiable.

We now state: „We propose the induced oxylipins as an second line of defense against K. algicida; in addition to the defensive proteases”

line 151: broken graph unjustifiable – show full curve

Since it is a logarithmic plot it would not be possible to plot it in an unbroken manner. The full curve is shown, no data has been left out.

line 155: control curve should appear superimposed on the figure

The graph shows the relative growth of K. algicida already normalized to the control without added HEPE. To superimpose the control would be confusing.

Reviewer 2 Report

Well designed work and well written. Keep going the study of algae resistance mechanisms!

At line 182, 78 rpm should be converted in X g. This allow anyone to easily execute the method.

Author Response

Reviewer 2

Comments and Suggestions for Authors

Well designed work and well written. Keep going the study of algae resistance mechanisms!

We thank the reviewer for the encouraging comment.

At line 182, 78 rpm should be converted in X g. This allow anyone to easily execute the method.

 In this case no centrifugation, but agitation of the culture is described. We clarify:

For recording the dose-response curve, K. algicida was cultivated in marine broth medium at 23°C and constant agitation on a shaker with 78 rpm.”

Reviewer 3 Report

The manuscript by Meyer et al. reports on studies of algal/diatom chemical ecology mediated by EPA-derived fatty acids. The work draws its conclusion from unsupported assumptions and weak correlations and needs significant re-focusing or additional supporting data.

A major concern is the leap from correlations of concentration variability to statements of causation. So three of seven (less than half) oxylipins (Fig 3) show statistically significant increase in bacterial-challenged diatoms relative to control diatoms. For those three, there is clearly a correlative effect, but no experimental evidence is presented as to what that effect is – defensive, offensive, or perhaps an unrelated downstream response. Instead, the authors take a metabolite that does not respond to the bacterial challenge and show that at concentrations (1 uM) well above what is found in the media (28 ng/mL = 88 nM)), the bacterial growth is slowed for six hours (but not longer). There is some discussion (lines 127-130) about phycosphere concentrations of the lipids and how they would exceed the effective concentration, but that's entirely conjecture and completely unknown. Even the authors state (line 126,127) '...it might very well contribute to the chemical defense...'. However, they move on into the discussion assuming it does, concluding 'The oxylipin therefore constitutes a first line of defense against...' the bacterium. That conclusion is not warranted based on the data provided.

Minor concerns:

line 20: 15-HEPE does not quantitatively inhibit K. algicida. It significantly inhibits it, but the relative growth (Fig 4) never falls below 0.2.

line 21-22: conclusion overstated

line 39: the term 'connected' has little scientific value. Do oxylipins cause wound-activated defenses or do they merely vary in response – those are both 'connections' but represent significantly different ecological implications

line 55-56: A discussion of how K. algicida, C. didymus and S. costatum interact in nature is important. Are they sympatric? Naive? And whatever they are, how does that matter for this study?

line 58: conclusion overstated

line 61: 'quantitatively' too strong a word

line 62: 'entirely' too strong

line 90: 'exceeds that of the growth' (drop 'curve')

Line 91: there is no growth curve in Fig 1 and 2

line 104: Fig 2 has no S. costatum data

line 107: 'but can arise alternatively by the abiotic oxidation of free fatty acids'. If this is true for S. constatum, why is this not true of C. didymus?

line 120: 'Consistently' is a weak scientific word. The level of HEPEs and 17-HDHA is not significant, so there is no correlation. No significance, no 'consistently'.

line 122: 12-HEPE did not increase significantly – the statement is wrong

line 124: RvE3 increased 'significantly' at the attamolar level – that needs to be put in context. Figure 3B should be part of Fig 3A with everything on the same scale. In the absence of bioassay data, an attamolar level increase in concentration is meaningless

line 132: broken graph unjustifiable. Put everything on scale.

line 139: that term 'connected' again. Not scientifically meaningful (see line 39 comment)

line 140: 15-HEPE was not significantly modulated by treatment with bacterial pathogen

line 143-144: intracellular concentrations are irrelevant because the bacterial pathogen never experiences intracellular conditions

line 145: 'connected' again

line 147: authors state 'This might be seen as an adaptation mechanism...' then conclude (line 149-150) that it is, without any support. Jumping from '...might be seen...' to '...therefore constitutes a first line of defense...' is unjustifiable.

line 151: broken graph unjustifiable – show full curve

line 155: control curve should appear superimposed on the figure

Author Response

The authors claimed to have identified oxylipins responsible of the resistance of C. didymus after bacterial infection with K. algicida. The authors described the production of oxylipins as induced by the bacterial infection.

The overall data are not so obvious, in Fig 2 only 5-HEPE seems to be induced but it is not the case in the Fig 3 after cell disruption. In this later figure, only the HEPE derivatives (18-HEPE and 11-HEPE but also RvE3) in very small amount appear to be modulated by the presence of bacteria. There is no evidence that these compounds were particularly active ones, the activity being tested on the major compound 15-HEPE.

See also comments to referee 1 – we have now disconnected the argument of induction from that of defense.

The evidence of 15-HEPE activity is also questionable. The estimate of the intracellular concentration is OK but is not related with the expected extracellular activity.

We have corrected this statement – we did not want to suggest that bacteria act within the diatom cells. Upon cellular disintegration the metabolites will be present in high local concentrations and act against the small bacterial cells.

 Furthermore, a bacterial resistance is proposed to be acquired in 6 hours and is not compatible with a protection of C. didymus over a 4 days period. A proposed explanation was the protease release from C. didymus Ref 26. In this reference, these proteases induced after 2 days might have an (additional) adverse effect on S. costatum.

In this manuscript we introduce a second line of defense in addition to the proteases. We agree that to argue about timelines would have to take into account too many unknown variables. This includes the timeline for expression of the protease (the 2 day value in the literature is a snahpshot). It also includes the local builing up of oxylipins around the cells and the question how bacteria associate to the algal cells. We now re-phrased the parts of the manuscript that could  be interpreted as kinetic discussion and focus on the observation of 1) induction and 2) activity.

The use of short term medium of C. didymus induced or not with K. algicida (before induction of proteases) should help the growing of S. costatum in presence of bacteria. This could constitute a good probe of the activity of the metabolites release by C. didymus. But it may be difficult to manage such experiment.

Given previous experiments that showed that medium of K. algicida is active against S. costatum we can expect that any K. algicida medium results in cell lysis of S. costatum. This is also supported by tripartite co-incubation experiments with S. costatum,  C. didymus and K. algicida that demonstrate universal activity against S. costatum independent of the presence of the co-culture partner (data are not shown here).

 Minor corrections:

L 104 The data concerning S. costatum are announced to be presented in the Fig 2 but absent!

Now “data not shown” is mentioned instead.

The amount of 17-HDHA seems different in Fig 3 when compared to Table 1 in Ref 10 (# 2,5 versus 1.86)

The order of magnitude is the same, and given the error bars (the error bars even overlap 2.5+-0.6  and 1.86+-0.18) the difference is negligible. Additionally: from the literature it is known that oxylipin production in diatoms is a highly variable parameter. Since data here and in Ref. 10 were obtained with different cultures in independent experiments it would not be justified to run a statistical test to evaluate if they are indeed statistically insignificantly different.

Round  2

Reviewer 1 Report

This manuscript "Induced algal oxylipins mediate the resistance of diatoms against algicidal bacteria" is submitted as an article in Marine Drugs.

The authors have modified the manuscript by introducing less pronounced statement.

However some inconstancies are still present. The title assume a complete effect of oxylipins whereas there is a better formulation is the abstract "may represent". By contrast the induction is clearly maintained in the abstract as well as line 60.

L 135 Why write increased if it is not significant.

The legend of Fig  4 was and is still wrong K. algicida should be read 15-HEPE.

 Overall, there is no strong evidence for induction of the production of oxylipins by K. algicida. The possible antibacterial activity was observed with only one derivative at a concentration of 1 µM.

The authors suggest two explanation i) the local concentration in the phycosphere ii) the sacrificial lysis of few cells to save the colony (lysis should induce a very strong dilution, the sentence L151-153 concluding a 10-fold the effective concentration is mysterious)

Both explanation are speculative without any real demonstration.

L155-157 The sentence is not based on the present results. 15-HEPE is not induced and the reality of the line of defense is not demonstrated.

 The manuscript is still a mixture of "induced" and "very efficient oxylipins" whereas most of the conclusions are only speculative.

Author Response

The authors have modified the manuscript by introducing less pronounced statement.

However some inconstancies are still present. The title assume a complete effect of oxylipins whereas there is a better formulation is the abstract "may represent". By contrast the induction is clearly maintained in the abstract as well as line 60.

We do not see the major problem of the reviewer. Results in Fig. 3 demonstrate an induced wound-activated production of some of the oxylipins. We therefore do not see the necessity to eliminate all references to induction of oxylipins in the manuscript. Since we have not tested the significantly induced metabolites for their activity but only the most abundant oxylipin (that is increased as well) we agree with the referee to avoid suggestion of an induced defense.

We adjusted these conclusion to reflect the fact.

Line 60ff :“We identify an induced up-regulation of a family of oxylipins in the wound activated response of the resistant alga as well as a release of these metabolites that may contribute to the chemical defense against the bacteria.”

In the abstract we now write “induces the production of several wound-activated oxylipins”

L 135 Why write increased if it is not significant.

We deleted the sentence

The legend of Fig  4 was and is still wrong K. algicida should be read 15-HEPE.

 Rephrased: Concentration-response curve for 15-HEPE inhibition of Kordia algicida growth

Overall, there is no strong evidence for induction of the production of oxylipins by K. algicida. The possible antibacterial activity was observed with only one derivative at a concentration of 1 µM.

See above, we show the results of induction in Fig. 4. We do not see why the referee insists to neglect these effects.

The authors suggest two explanation i) the local concentration in the phycosphere ii) the sacrificial lysis of few cells to save the colony (lysis should induce a very strong dilution, the sentence L151-153 concluding a 10-fold the effective concentration is mysterious)

Both explanations are speculative without any real demonstration.

The concept of the phycosphere is a generally accepted one (and we cite well accepted reviews for this). The fact that high concentrations will be present after lysis is obvious. Clearly the lack of appropriate single cell analysis given by today´s instrumentational limits makes the statement speculative but we phrased it as such.

 In response we adjusted the conclusion to be entirely precise:

We here extend the concept of induced oxylipin-production to the field of C20 oxylipins. We suggest that these metabolites might be involved in the chemical defense to diatoms.

L155-157 The sentence is not based on the present results. 15-HEPE is not induced and the reality of the line of defense is not demonstrated.

This sentence has been rephrased to

We propose the C20 oxylipins could represent a line of defense against K. algicida that acts in addition to the previously reported defensive proteases [18].

 The manuscript is still a mixture of "induced" and "very efficient oxylipins" whereas most of the conclusions are only speculative.

We do not use the term efficient or very efficient in the manuscript. We verified that after the adjustments made above the term induced is mentioned only in the context of Fig. 4 not of a defense.